# Electrodischarge Drilling of Microholes in c-BN

**DOI:** 10.3390/mi11020179

**Published:** 2020-02-10

**Authors:** Dominik Wyszynski, Wojciech Bizon, Krzysztof Miernik

**Affiliations:** 1Faculty of Mechanical Engineering, Cracow University of Technology, al. Jana Pawla II 37, 31-864 Krakow, Poland; wojciech.bizon@pk.edu.pl; 2Faculty of Materials Science and Physics, Cracow University of Technology, al. Jana Pawla II 37, 31-864 Krakow, Poland; kmiernik@pk.edu.pl

**Keywords:** electrodischarge micromachining, drilling, cubic boron nitride

## Abstract

Cubic boron nitride (c-BN) is a “difficult-to-cut” material. High precision machining of this material is problematic because it is difficult to control the material removal rate and maintain acceptable accuracy. This paper describes an application of electrodischarge machining (EDM) for drilling micro holes in c-BN. The goal of this research was to determine a set of parameters and technical specifications for such a process. We used an isoenergetic transistor power supply with a microsecond voltage pulse generator and a tungsten tool electrode of diameter d = 381 μm. Each hole was drilled for 10 min. The holes did not exceed 410 μm in diameter and were at least 1000 μm deep. The process was carried out in a hydrocarbon dielectric liquid. We assess the quality of the holes from a qualitative and quantitative point of view. The results show that electrodischarge is a precise, accurate, and efficient method for machining c-BN.

## 1. Introduction

Machining of very hard and high strength materials—called “difficult-to-cut” materials—has been a challenge for production engineers for decades [1]. The low efficiency of current machining processes and expensive machine tools make precise industrial-scale machining for this category of materials expensive. Boron nitride (BN) is among these difficult-to-cut materials and, due to its extraordinary properties, has been present in various technical applications for more than 150 years. These properties vary depending on the polymorph structure of BN. This chemical compound exists in an amorphous form (a-BN) and in its basic and the most stable soft hexagonal form (h-BN), which is commonly used as a lubricant. The cubic form of boron nitride c-BN, however, is one of the hardest materials on Earth [2]. Its hardness makes it very attractive in many applications, and c-BN is used more frequently than the other forms of boron nitride. Due to high thermal and chemical stability, c-BN is widely used in the manufacturing of cutting tools for ferrous alloys machining, where diamond tools are less durable due to carbonization and chemical solubility. There is also a wurtzite form of BN, which is structurally similar to c-BN. It is said to be 18% harder than diamond, but due to its rare occurrence in nature, this has not been scientifically verified.

Even if recent developments in material science have introduced an efficient way to create c-BN, which makes the material more viable for applications such as high-power electronics, transistors, and solid-state devices, they have not resolved the problem of the inefficient machining of c-BN [3]. The most frequently used technology for manufacturing parts of c-BN is sintering [4,5]. This process introduces some limitations, i.e., limited part size and complexity (internal curvilinear channels) as well as the high cost of tooling. The novelty of this research is to apply electrodischarge machining on a micro-scale in order to offer a cost-effective and versatile approach to forming c-BN parts. It would make it more attractive for a broad variety of engineering applications where expensive and limited (in terms of depth of the holes) laser drilling methods are applied.

## 2. Materials and Methods

Machining of difficult-to-cut materials such a c-BN requires non-traditional machining processes. These methods are considered unconventional because they do not require direct contact of the tool with the machined material. The energy necessary to remove the machined material is delivered by means of kinetics (i.e., abrasive water jet and ultrasound abrasive machining), electromagnetic radiation (i.e., laser beam machining [6]), or electric field (i.e., electrochemical and electrodischarge machining [7]).

In order to drill with acceptable accuracy and precision, we used electrodischarge machining which is effective given the partial electroconductivity of c-BN. EDM is an electrically induced thermal process, whereby the machined material is removed from the workpiece by energy from electrical discharges occurring between the working electrode tool and the workpiece electrode. The electrodes are immersed in a dielectric medium (air, deionized water, hydrocarbon liquids, etc.) Both the workpiece and the electrode tool material are removed by melting and evaporation coming from energy generated by electrical discharges or sparks in the inter-electrode gap. The role of the dielectric is to provide optimal conditions (heat exchange and flow) for discharge and to evacuate debris from the inter-electrode gap between the voltage pulses [8,9]. Figure 1 below presents a scheme of the electrodischarge process using a tubular electrode tool. Rotation is introduced to better clean the debris (eroded particles) from the inter-electrode gap.

The current research was motivated by needs voiced by manufacturers of cutting tools for aircraft parts’ machining. The objective was to check the feasibility of the application of the EDM method for sinking or drilling channels in the c-BN layer of an insert for grooving, and to compare it to laser machining. The results of laser machining were not included in the current research.

For the purpose of the research we used a Sandvik Coromant CB20 grade cutting tool as a machined part to make blind holes in the c-BN layer by means of micro EDM drilling (see Figure 2).

We chose this cutting tool because it allowed us to deliver various cutting fluids directly to the machining zone (under the tip), which improved cutting efficiency. To this end, the top surface (black) of the insert was subjected to several electrodischarge drilling tests. The experimental part was preceded by an analysis of the authors’ experience in electrodischarge machining of difficult-to-cut materials, preliminary machining tests, and confirmed with the results presented in [6]. Based on these preliminary tests, the range of the most important parameters was selected and the experiment was planned. The experiment was prepared in accordance with factorial design [12]. After preliminary tests, we have decided that the experiment plan should cover a relatively wide range of pulse-on time and symmetrical pulse-off durations (1 and 10 µs). The goal was to check the process indices for the shortest possible, pulse-on times and relatively longer ones on our pulse generator. We assumed a maximum of 10 µs pulse-on times in order to not overheat the inter-electrode gap that could result in excessive electrode tool wear and dielectric decomposition to graphite. The excessive appearance of conductive graphite corrupts the machining process. Detailed information about the design of the experiment is presented below in Table 1, Table 2 and Table 3. Table 4 below presents output process factors and measures.

The following aspects of the process were measured and determined:

The threshold current (I_t_) is the minimum current value acceptable for feed regulator that should be maintained to support the discharges, while the working current (I_w_) is the current value that is a reference for the feed regulator to be the default current during the drilling.

The holes were drilled with the micromachining machine prototype designed and built in the Institute of Production Engineering at the Cracow University of Technology in Krakow, Poland, presented below in Figure 3. The machine body was designed and manufactured of materials ensuring minimal thermal expansion and high stiffness (granite).

This hybrid micromachining machine prototype was designed and manufactured for micromachining involving pulse electrochemical machining and electrodischarge machining. The application of both the aforementioned methods in a sequential or synergic way enables obtaining most of the advantages of both methods. For the current study, the machine tool was used in the EDM work regime and equipped with a transistor isoenergetic voltage pulse generator and a power supply that enables setting rectangular voltage pulses at a range from 1 to 999 μs and an amplitude of 60–120 V. The chosen cylindrical tungsten electrode tool, which is produced by Balzer Technik in Switzerland, of ϕ = 381 was clamped on a Sarix, Switzerland clamping tool. The sample was fixed with an EROWA ITS 50, Switzerland clamping tool. As the working electrode tool wears during machining due to electrical discharges, we used a high melting point T = 3410 °C tungsten electrode tool [14]. The phase diagram for c-BN, presented in Figure 4, shows that the temperature required to melt or evaporate the machined material is relatively high (more than 3000 °C). Application of the standard low melting point copper electrode could result in excessive electrode tool wear. Moreover, the tungsten electrode tool’s Young modulus is higher and the electrode tool is less prone to plastic deformation during fixing in the clamping tool and homing. Unfortunately, no tungsten tubular electrode tool of this diameter is commercially available. Preliminary machining tests revealed also excessive tungsten electrode tool wear. In the current research positive (higher) electrical potential was applied to the machined part. The chosen polarity of the electrodes ensures maximal material removal rate and minimal electrode tool wear.

Initially, the electrode tool was not rotated during the sinking process (first two holes). Then the electrode tool was rotated for drilling in order to improve the removal of resolidified electroerosion products from the inter-electrode gap. The gap was flushed with fresh dielectric from the side. A scheme of the test stand is presented below in Figure 5.

## 3. Results and Discussion

The goal of the work was to describe the possibility of application of the method for machining of cubic boron nitride. The research was designed and prepared to show the potential of the method and describe technological aspects. The measurements were taken with the use of an optical microscope Motic series K equipped with Instant Digital Microscopy camera Moticam 2300 (1/2" Live 3.0 Megapixels, Hongkong, China), and Motic Images Plus software (Hongkong, China). The scanning electron microscopy images were prepared by JEOL JSM-5500 Scanning Electron Microscope (Tokyo, Japan).

Two magnification levels (200× and 1000×) are displayed in order to give a comprehensive view of the machined holes’ shape and edge quality. The noticeable shape inaccuracy (Figure 6a) was caused by carbon (graphite) deposited on the electrode tool surface. Relatively long voltage pulse (t_on_ = 10 µs), high working current value (I_w_ = 0.9 A), and insufficient dielectric flush in the inter-electrode gap caused hydrocarbon dielectric thermal decomposition and graphite deposition on the surface of the electrode tool. The electrically conductive graphite took over the role of the tungsten electrode tool at the deposited area. Nevertheless, the shape of the electrode tool was acceptably copied, which can be observed in the round shape of the hole in Figure 6. It would be difficult to measure the edge surface roughness Ra precisely, but it was estimated by digital image analysis to be lower than Ra < 5 μm. The Ra was evaluated based on known magnification of SEM (Scanning Electron Microscope) image and proportion of used marker. For example, if the 10 µm marker has 50 pixels, then counting the size of spatial amplitudes (of the valleys and peaks on the edge of the hole) in pixels gives rough information about the physical size and enables to evaluate Ra upon a mathematical formula. Digital image analysis can give approximated values of 2D roughness. The same phenomenon related to inter-electrode gap overheating and dielectric thermal decomposition was observed in the second sinking approach. This inaccuracy was eliminated in successive drilling tests by more intense dielectric flushing [16] and electrode tool rotation. The hole remained round and the edge sharp. The estimated surface roughness Ra was less than 10 μm.

Intensified dielectric flushing improved evacuation of the debris from the inter-electrode side gap (see Figure 7a). The graphite from high-temperature dielectric decomposition was not deposited on the electrode tool, and there was no deformation. The shape was properly reproduced from the cylindrical electrode tool.

Figure 7 shows that, from the qualitative point of view, the shape of the hole is round. The SEM pictures of all obtained holes confirm this tendency. It would also be interesting to know the chemical composition of the hole edges. Unfortunately, energy-dispersive X-ray spectroscopy could not determine the chemical composition because the atomic mass of the examined compound was too small. Nevertheless, the SEM images (Figure 6 and Figure 7) showed no visual changes of the material surface on the edge. This is consistent with limited heat-induced phase change in this area.

The depth of the holes was calculated with a stepper motor encoder. Reading values were reduced by the linear electrode tool wear and inter-electrode gap thickness.

The measured and calculated quantities are presented in Figure 8 below and summarized in Table 5.

Figure 8 shows that the pulse-on and off times have a significant impact on hole depth for higher working current values while the impact of the increase in hole diameter could be neglected. The electrical current intensity is proportional to the number of charges that are transferred in a given time. As electrical potential, in this case, is constant and the power P = U × I, the bigger the I value, the bigger the P value. P stands for the energy (work) necessary for material melting and evaporation over time of machining. As the machining time, in this case, is constant, the higher the P value, the deeper the hole. The energy in this case is consumed for drilling (in the z-axis direction) due to the feed regulator that compensates inter-electrode gap thickness and thus enables discharges. The side gap thickness is constant due to continuous working electrode feed and its circumferential wear. When the working current I_w_ is higher, higher linear electrode wear and hole diameters are also observed. It leads to a higher side gap size that can facilitate the removal of debris from the inter-electrode volume during the pulse-off time. This can improve electrical erosion conditions. Figure 9 shows the relation between average drilling speed and changes of working current I_w_ for various pulse-on and pulse-off times. Average drilling speed relates to the depth of the hole over the machining time. One can observe that the higher working current has an impact on average drilling speed. Therefore, it improves the material removal rate. The relation between working current I_w_ and MRR is presented in Figure 10.

Designer and metrology communities face a huge challenge regarding the measurement of micro-scale features [17]. Out-of-roundness depends on the size and is defined in so-called International Tolerances. If we take these norms into consideration, according to standard tolerance grades, our results fit IT8-IT9. This result needs to be explained. These values of tolerances refer to features ranging from 0 to 3150 mm in size. The dimensions of the holes prepared in the research are close to zero (micrometer scale). It is worth considering if the dimensions of holes presented in the current research are not too close to the tolerance value. For that reason we propose the formula for determining relative out-of-roundness in Equation (1) below in order to quantitatively address the roundness of the holes:(1)$$\mathrm{RoR}=\frac{\mathrm{Maximum}\;\mathrm{inner}\;\mathrm{diameter}-\mathrm{Minimum}\;\mathrm{inner}\;\mathrm{diameter}}{\mathrm{Nominal}\;\mathrm{diameter}}\times 100\%$$
where:

maximum inner diameter = mean hole diameter + 1SD,

minimum hole diameter = mean hole diameter - 1SD,

nominal diameter = mean hole diameter.

Relative electrode tool wear refers to the linear electrode tool wear in regard to the depths of the holes.

As the prepared holes were of small diameter (ca. 400 µm), of a relatively high depth (ca. 1000–1300 µm), and were blind holes, it is difficult to present the sidewall profile. It could be assumed that the surface roughness value is close to the one determined on the edge of the hole. The linear electrode tool wear was dependent on the depth of the hole and voltage pulse energy. The linear wear was measured and presented in Table 5 above (less than 200 µm). The circumference electrode tool wear is definitely low and dependent on the time that the electrode tool spent in the material during machining. In the area close to the face of the electrode tool the circumferential wear is usually higher than in the area close to the hole edge due to machining time. It has some impact on the hole diameter decreasing along the depth profile. The determined side gap value (the lowest for the less energetic pulses ca. 20–30 µm) allows assuming that the circumferential wear value is not high and does not introduce significant taper of the holes.

The best set of parameters for drilling are those that balance higher material removal rate, lower electrode tool wear, and smaller side gaps, and is the one that ensures good shape and accuracy and cost-effectiveness. Of those presented above in Table 1, Table 2 and Table 3, the best conditions for micro EDM drilling were achieved when the working current was the highest. Relatively high working current and short pulses improved machining quality. The resulting increase in hole diameter was negligible, while the material removal rate was significantly higher and the electrode tool wear was acceptable.

## 4. Conclusions

The tests prepared in the study proved that the micro EDM drilling process is stable, reliable and efficient. The obtained results allow formulating the conclusion that electrodischarge drilling offers c-BN tools’ manufacturers a cheap, accurate, and precise alternative to other machining methods. Micro EDM drilling enables the manufacturing of channels for delivering various cutting fluids directly to the machining zone (under the tip) to improve cutting efficiency and extend the tip life of cutting tools [17]. Designer and metrology communities face a huge challenge regarding the measurement of micro-scale features [18]. The analysis of the surface geometrical structure features of the holes obtained in current research revealed that there is still a need to bridge the gap in measurement standards concerning the so-called mezzo scale (transient between millimeter and micrometer scale). Nevertheless, it could be concluded that micro electrodischarge drilling of c-BN can also be applied in many manufacturing applications i.e., semiconductor devices for harsh environments such as solar-blind UV sensors in space [19]. Due to c-BN’s extraordinary thermal conductivity, it can also be applied in the manufacturing of heat sinks for semiconductor lasers and microwave devices. The resolution of micro electrodischarge drilling depends on minimal removed volumes. In order to improve the machining accuracy and precision, it is very important to apply adequate machine body design and materials limiting temperature expansion and ensuring its required stiffness. The other important factors are power supply, voltage pulse generators, as well as the properties of electrode tool materials and reliability of clamping systems. Also, the machined material homogeneity has significant importance for the obtained results. Of course, this approach is adequate for finishing operations or for machining on a micro-scale. This research was part of the Innolot project “Technologies of forming micro- and macro-geometry of the cutting tools, made of ultra-hard materials in the process of implementation of advanced laser techniques” founded by the Polish National Centre for Research and Development.

## Figures and Tables

**Figure 1 micromachines-11-00179-f001:**
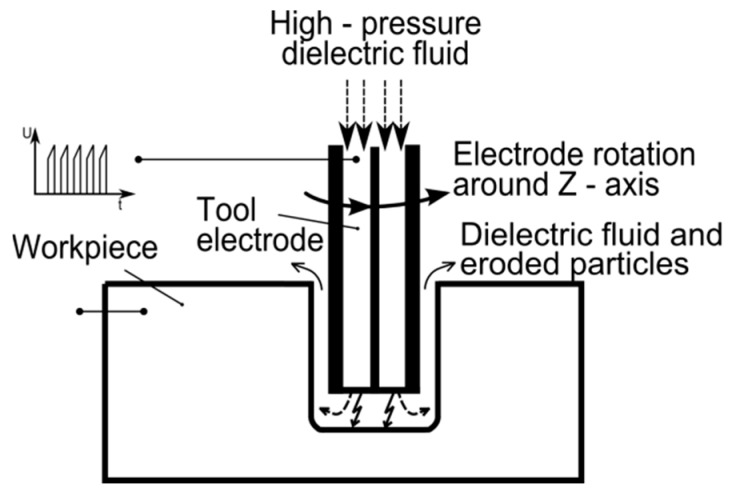
Scheme of EDM drilling [10].

**Figure 2 micromachines-11-00179-f002:**
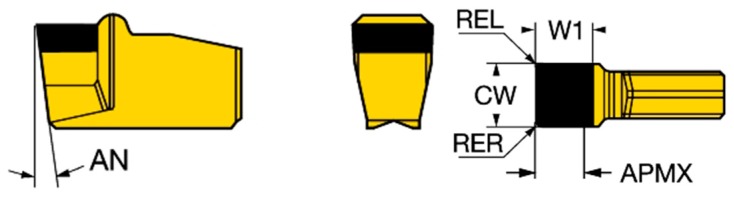
T-Max^Ⓡ^ Q-Cut insert for grooving (N151.2-600-50E-G CB20) [11].

**Figure 3 micromachines-11-00179-f003:**
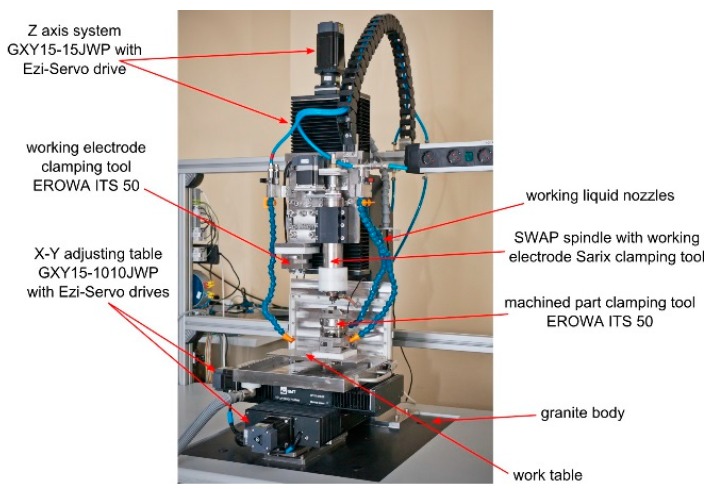
Electrochemical/electrodischarge hybrid micromachining machine prototype [13].

**Figure 4 micromachines-11-00179-f004:**
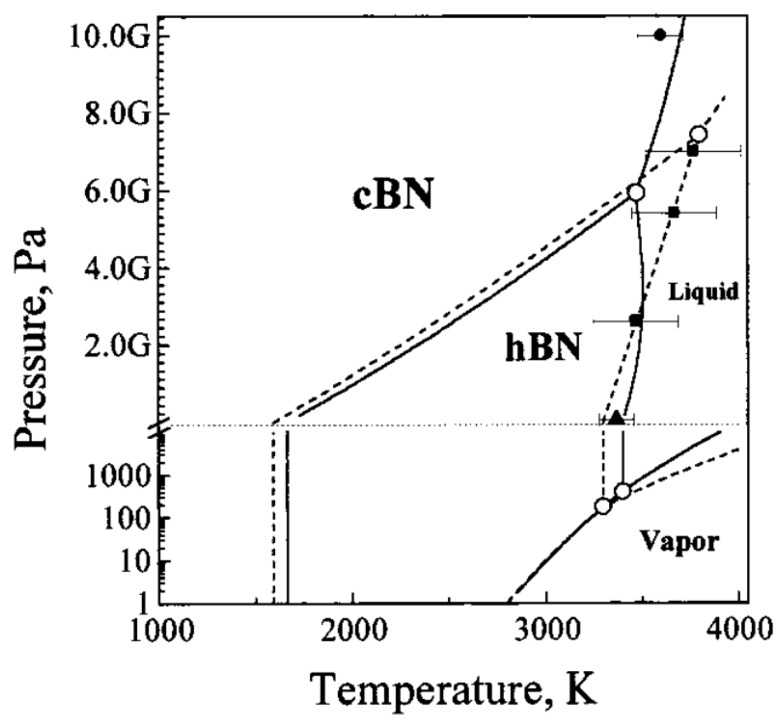
Phase p, T-diagram of boron nitride [15].

**Figure 5 micromachines-11-00179-f005:**
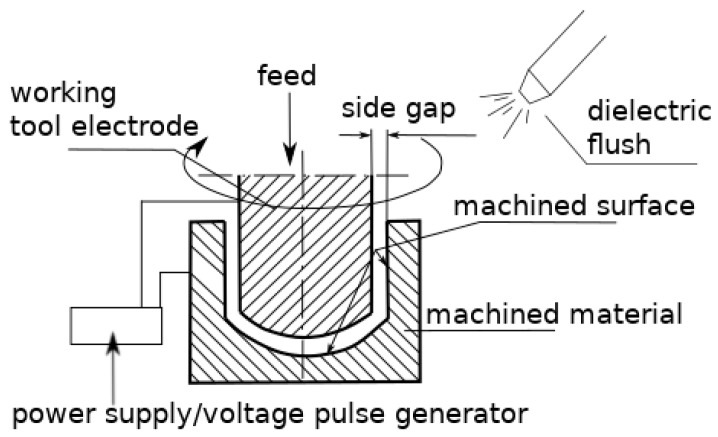
Scheme of the electrodischarge drilling process.

**Figure 6 micromachines-11-00179-f006:**
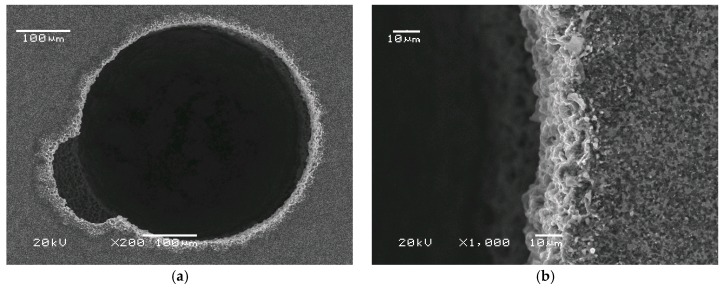
SEM images of hole no.1. U = 120V, t_on_ = 10 µs, t_off_ = 10 µs, threshold current I_t_ = 0.3 A, working current I_w_ = 0.9 A. (**a**) magnification 200×, (**b)** magnification 1000×.

**Figure 7 micromachines-11-00179-f007:**
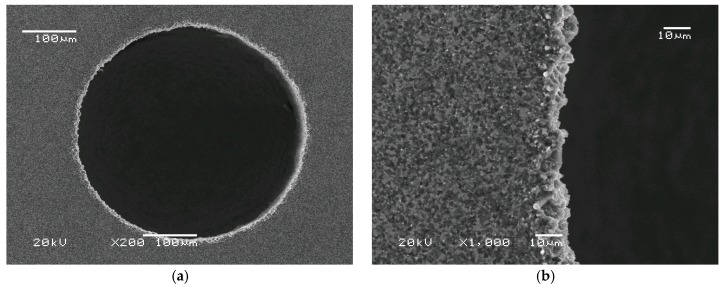
SEM images of hole no.4. U = 120 V, t_on_ = 1 µs, t_off_ = 1 µs, threshold current I_t_ = 0.3 A, working current I_w_ = 0.9 A. (**a**) magnification 200×, (**b**) magnification 1000×.

**Figure 8 micromachines-11-00179-f008:**
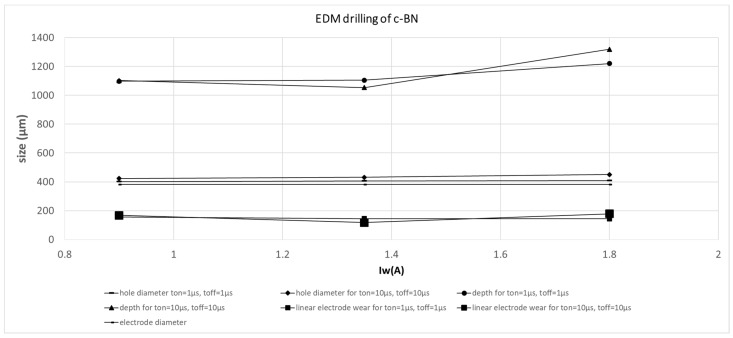
Results of electrodischarge drilling with the use of a tungsten electrode tool in a c-BN sample for various t_on_ and t_off_ values.

**Figure 9 micromachines-11-00179-f009:**
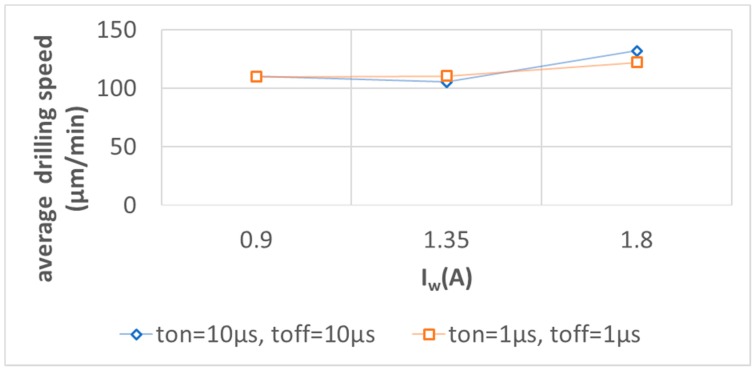
Average electrodischarge drilling speed for various t_on_ and t_off_ and working current I_w_.

**Figure 10 micromachines-11-00179-f010:**
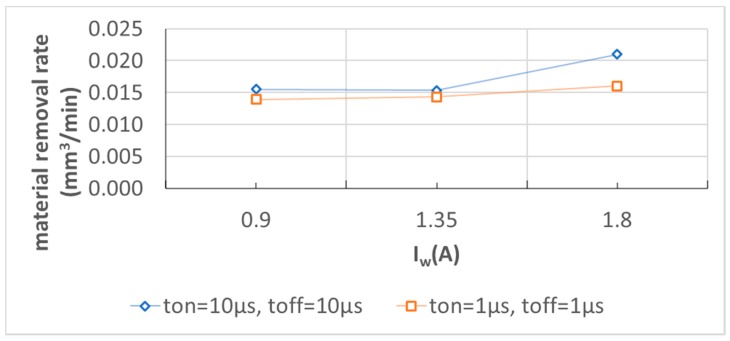
The material removal rate for various t_on_ and t_off_ and working current I_w_.

**Table 1 micromachines-11-00179-t001:** Input factors for the experiment.

Factors	Parameters	Values
Constant factors:	Voltage (V)	120
	Machining time (s)	600
	Material and diameter of electrode tool (μm)	Tungsten, ϕ = 381
	Dielectric liquid	Exxol80 (hydrocarbon)
	Electrode tool rotation (rpm)*	250
	Pulse duty cycle D (%) $D=\frac{t_{on}}{t_{on}+t_{off}}\cdot 100\%$	50
	Threshold current I_t_ (A)	0.3
Disrupting factors:	Uneven dielectric liquid flow	
Variable process parameters:	Pulse-on time t_on_ (μs)	1; 10
	Pulse-off time t_off_ (μs)	1; 10
	Pulse period (μs)	2; 20
	Pulse frequency (kHz)	500; 50
	Working current I_w_ (A)	0.9, 1.35, 1.8

**Table 2 micromachines-11-00179-t002:** The research plan—series one.

No.	Constant Values			Variable Parameter
	t_on_ (μs)	t_off_ (μs)	I_t_ (A)	I_w_ (A)
1	10	10	0.3	0.9
2	10	10	0.3	1.35
3	10	10	0.3	1.8

**Table 3 micromachines-11-00179-t003:** The research plan—series two.

No.	Constant Values			Variable Parameter
	t_on_ (μ)	t_off_ (μs)	I_t_ (A)	I_w_ (A)
4	1	1	0.3	0.9
5	1	1	0.3	1.35
6	1	1	0.3	1.8

**Table 4 micromachines-11-00179-t004:** Output process factors and measures.

Processing Factors
Hole diameter (μm)
Hole depth (μm)
Linear electrode tool wear (μm)
Average drilling speed (μm/min)
Material Removal Rate (μm^3^/min)

**Table 5 micromachines-11-00179-t005:** Mean and corrected standard deviations (SD) of the measured and calculated values (Bessel’s correction).

Hole Number	Hole Diameter (μm) SD = 10	Hole Depth (μm) SD = 74	Linear Electrode Wear (μm) SD = 3	Average Drilling Speed (μm/min) SD = 7	Relative Electrode Wear (%) SD = 0.3	Side Gap (μm) SD = 10	MRR (mm^3^/min) SD = 0.002	Edge Ra (μm)	Relative Roundness
1	423	1102	167	110	15	43	0.016	< 5	4.7%
2	431	1053	118	105	11	50	0.015	< 10	4.6%
3	450	1319	178	132	13	69	0.021	< 10	4.4%
4	402	1097	157	110	14	21	0.014	< 5	4.9%
5	406	1105	145	111	13	25	0.014	< 5	4.9%
6	409	1220	144	122	12	27	0.016	< 5	4.8%

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
