# Peer review of "Electrodischarge Drilling of Microholes in c-BN"

_micromachines, 2020, doi:10.3390/mi11020179_

Round 1

Reviewer 1 Report

The manuscript reports on micro electrodischarge drilling of c-BN inserts.

The reviewer find the manuscripts of interest, however, few comments to improve the paper and strengthen its argument are listed below.

The literature review needs extension to include more related work such as different machining techniques (such as laser ablation and EDM) used to machine c-BN.  The list of references contained outdated very old Refs, please update. The DoE along with the applied process parameters need further clarification and justification. The results and Discussion section needs improvement to properly explain the results. What is reported is only an observation of the results without sufficient explanation of the effect of applied process parameters on the obtained results. the conclusion are too long, tedious and contains unnecessary details. Please revise to condense and highlight the significant and generic findings of this research study. 

Author Response

Dear Reviewer,

Thank you very much for your effort and constrictive remarks concerning our manuscript quality. We appreciate your valuable knowledge and experience. We believe our manuscript would be more interesting and clear to the Readers due to your accurate comments. Kindly please find the answers for your notices below.

Dominik Wyszyński

Reviewer 1

The manuscript reports on micro electrodischarge drilling of c-BN inserts.

The reviewer find the manuscripts of interest, however, few comments to improve the paper and strengthen its argument are listed below.

The literature review needs extension to include more related work such as different machining techniques (such as laser ablation and EDM) used to machine c-BN.

Ad. 1. Information on other machining methods was complemented:

Denkena, A. Krödela), T. Grove, On the pulsed laser ablation of polycrystalline cubic boron nitride—Influence of pulse duration and material properties on ablation characteristics. Journal of Laser Applications, Vol. 31, Issue 2, March 2019.

Jia, Y.H., 2011. Study on EDM Technics of Polycrystalline Cubic Boron Nitride Cutting Tool and PcBN Cutting Tool’s Life. Applied Mechanics and Materials 120, 311–315. https://doi.org/10.4028/www.scientific.net/amm.120.311

The list of references contained outdated very old Refs, please update.

Ad. 2. The reference list was updated.

The DoE along with the applied process parameters need further clarification and justification.

Ad. 3 Design of experiment was clarified and justified the reference to the handbook [12] was added. All changes are marked yellow in the corresponding section of the manuscript.

The results and Discussion section needs improvement to properly explain the results. What is reported is only an observation of the results without sufficient explanation of the effect of applied process parameters on the obtained results. the conclusion are too long, tedious and contains unnecessary details.

Ad. 4. The results explanation and conclusions were improved. Changes were yellow highlighted.

Reviewer 2 Report

Review of micromachines-704627-peer-review-v1: “Electrodischarge drilling of microholes in c-BN”

The subject of the paper is relevant with the topics of the journal, and offers a great deal of industrial value.

It would improve the quality of the paper if the authors were willing to incorporate the following:

Line 85 and ref 10 deal with a micromachining machine prototype designed and built in the 84 Institute of Production Engineering. A paragraph describing this machine would help the reader of this paper. 8. larger size of the description of the experiments and the symbols would be great. The authors should add the numbers together with the graph. 9 and 10. Should be larger in order to be clear to the reader. The authors should add the numbers together with the graph.

My proposal to the editor is to accept the paper after minor revisions.

Author Response

Dear Reviewer,

Thank you very much for your effort and constrictive remarks concerning our manuscript quality. We appreciate your valuable knowledge and experience. We believe our manuscript would be more interesting and clear to the Readers due to your accurate comments. Kindly please find the answers for your notices below.

Dominik Wyszyński

Reviewer 2

Review of micromachines-704627-peer-review-v1: “Electrodischarge drilling of microholes in c-BN”

The subject of the paper is relevant with the topics of the journal, and offers a great deal of industrial value. It would improve the quality of the paper if the authors were willing to incorporate the following:

Line 85 and ref 10 deal with a micromachining machine prototype designed and built in the 84 Institute of Production Engineering. A paragraph describing this machine would help the reader of this paper.

Ad. 1. The paragraph with description of micromachining machine prototype was added. Lines 90-94.

Larger size of the description of the experiments and the symbols would be great. The authors should add the numbers together with the graph. 9 and 10. Should be larger in order to be clear to the reader. The authors should add the numbers together with the graph.

Ad. 2. The corrections were included.

Reviewer 3 Report

The paper is well written and it appeals at first glance; unfortunately, the content of the manuscript doesn't keep the lines promoted in the abstract. The state of the art doesn't help to understand what the novelty of the proposed work actually is. Furthermore, the novelty is by no means clear throughout the paper. The organization of the paper is somehow caotic, it lacks logic and effective building structure, it sounds very shallow. The experimental settings is not thouroughly described (important process parameters are not indicated). The choice of the performance indexes are debatable, or at least not clearly defined and estimated. Several results are not discussed. For all these reasons, I have to reject the paper.

I strongly invite the auhtors to consider very carefully some comments reported below, provided to help them improve their paper prior any other re-subimission to this journal.

Comments:

Please insert “-“ among the words: difficult-to-cut

Line 39: replace “a efficient way” with “an efficient way”

Line 42-43: Concerning sintering of c-BN, the authors stated that: “it introduces limitations like limited part size and complexity as well as high cost of tooling”. What do the authors mean by writing with “limited part size and complexity”?

Line 43: the following statement “apply electrodischarge machining on a micro scale in order to offer a cost effective and versatile approach to forming c-BN parts”. In particular: please justify in brief how micro-EDM can be cost effective and versatile approach to forming c-BN parts.

Section 2: please, rewrite “workpiece” as whole word

Line 58: melting and evaporation, not “or”

In table 1, the information regarding pulse frequency is not reported. Please, include it in the table. Additionally, the authors should provide a justification about the choice they made for the selected process parameters. For instance: Did these parameters provide better performance (MRR, TWR and so on)? Or were these parameters already tested for the machining of c-BN and obtained by previous analysis? If yes where (state of the art or personal investigation9? Please, clarify this point.

Dealing with drilling of micro-holes, the taper of the holes is generally considered: how come the authors did not consider this parameter in the evaluation of drilling performance?

It is not very clear what the working current Iw is for the authors? Is it the current discharged in the interelectrode gap? Is it the maximum expected current related to the generated pulse? Please clarify.

What kind of pulse generator the authors used in the experiments? Transistor or relaxation type? It is important to understand the micro-EDM drilling pulse shape when estimating the performance. Unfortunately, by saying “rectangular voltage pulses” the authors did not help in clarifying this important aspect of the machining. 

Line 101-102: the authors stated “The chosen polarity of the electrodes ensure maximal material removal rate 101 and minimal electrode tool wear.” What is the polarity of the tool? It is supposed to be negative, but the authors must indicate it clearly.

Line 105: Why was the tool not in rotation during the machining of the first Two holes?

The quality of Figure 8 is very low: it is very difficult to read the legends and as such to estimate the plots. It is not clear what the x-axis refers to.

Line 147 states “in figure 8 the pulse time on and off has…”. please rephrase the sentence “in figure 8, pulse on time and pulse off time have…”.

Concerning figure 10: A mistake is present in the title of figure related to MRR (pulse on/off vs MMR). Same mistake is present in Table 5.

Figure 9 and 10 are not commented in the manuscript.

Moreover, the authors should explain from a physical viewpoint the following statement:

-  “toff and ton have significant impact on hole depth for higher working current values…”.

- “…while the increase in hole diameter is negligible”.

From line 149, the authors came back discussing SEM images. This organization is quite confusing. Please, take care of the manuscript structure!

In line 168, the authors finally stated that the micro-holes are blind. I warmly suggest to provide this information in the very beginning of the manuscript (title, abstract and so on).

Line 169: the authors mentioned tool wear, but no measurements or calculations have been done throughout the entire manuscript. This is quite confusing.

In line 170, it is stated “the linear wear was measured and presented in the table 5”. It is by no means clear what the authors referred to. Tool wear and linear wear are not the same thing. I strongly suggest to include a definition of the quantity the authors dealt with. In literature, tool wear is often indicated as "volume" tool wear, whereas the linear tool wear refers to the shortening of the tool, assuming that the tool section is kept constant during a micro-EDM process. Nonetheless, micro-EDM drilling of micro-holes is far acknowledged as a process whose tool wear is dramatically outstanding, especially concerning the tool section. It is not just the tool shortening (linear tool wear) but the section variation that matters most, since it introduced the hole tapering. The hole tapering is particularly important in high aspect ratio holes and, although it cannot be easily measured in case of blind holes, it should be extrapolated by the observation of the volume tool wear.

Author Response

Dear Reviewer,

Thank you very much for your effort and constrictive remarks concerning our manuscript quality. We appreciate your valuable knowledge and experience. We believe our manuscript would be more interesting and clear to the Readers due to your accurate comments. Kindly please find the answers for your notices below.

Dominik Wyszynski

Reviewer 3.

The paper is well written and it appeals at first glance; unfortunately, the content of the manuscript doesn't keep the lines promoted in the abstract. The state of the art doesn't help to understand what the novelty of the proposed work actually is. Furthermore, the novelty is by no means clear throughout the paper. The organization of the paper is somehow caotic, it lacks logic and effective building structure, it sounds very shallow. The experimental settings is not thouroughly described (important process parameters are not indicated). The choice of the performance indexes are debatable, or at least not clearly defined and estimated. Several results are not discussed. For all these reasons, I have to reject the paper.

I strongly invite the auhtors to consider very carefully some comments reported below, provided to help them improve their paper prior any other re-subimission to this journal.

Comments:

Please insert “-“ among the words: difficult-to-cut.

Ad. 1. Hyphen was added.

Line 39: replace “a efficient way” with “an efficient way”

Ad. 2. Correction was made.

Line 42-43: Concerning sintering of c-BN, the authors stated that: “it introduces limitations like limited part size and complexity as well as high cost of tooling”. What do the authors mean by writing with “limited part size and complexity”?

Ad. 3. The authors mean by this that in example that it would be difficult to sinter the c-BN parts with internal channels in example.

Line 43: the following statement “apply electrodischarge machining on a micro scale in order to offer a cost effective and versatile approach to forming c-BN parts”. In particular: please justify in brief how micro-EDM can be cost effective and versatile approach to forming c-BN parts.

Ad. 4. The sentence concerns particularly manufacturing parts (tools) with micro size diameter holes and fluid delivery channels.

Section 2: please, rewrite “workpiece” as whole word.

Ad. 5. Correction was applied.

Line 58: melting and evaporation, not “or”

Ad. 6. Authors are not sure what exactly are temperature and pressure in the inter-electrode gap during the discharge and plasma channel formation. The phase transitions (solid to liquid and liquid to vapour) in the inter-electrode gap are difficult to describe due to sudden and impulsive nature of the phenomena occurring there. Also the phase diagram does not address the answer clearly. Nevertheless Authors trust in the Reviewer’s experience and introduce the change.

In table 1, the information regarding pulse frequency is not reported. Please, include it in the table.

Ad. 7. In Authors opinion the pulse frequency is not that much significant as the pulse time on and pulse time off and the duty cycle. Nevertheless the pulse period and frequency  was included for the Reviewer’s convenience.

Additionally, the authors should provide a justification about the choice they made for the selected process parameters. For instance: Did these parameters provide better performance (MRR, TWR and so on)? Or were these parameters already tested for the machining of c-BN and obtained by previous analysis? If yes where (state of the art or personal investigation9? Please, clarify this point.

Ad. 8. The presented in Figures 8 -10 results of MMR and linear electrode tool wear are standard EDM drilling process indicators. The value of shortening of the electrode is an important information from the drilling process planning. It is possible to evaluate TWR and thus compensate the impact of electrode tool wear in the drilled hole depth. The analysis of the presented results gives information that the best set of parameters for drilling are those that balance higher material removal rate, lower electrode tool wear and smaller side gaps and is the one that ensures good shape and accuracy and is also cost-effective. Of those presented in Tables 1 - 3 , the best conditions for microEDM drilling were achieved when the working current was the highest. Relatively high working current and short pulses improved machining quality. The resulting increase in hole diameter was negligible, while the material removal rate was significantly higher and electrode tool wear was acceptable.

Dealing with drilling of micro-holes, the taper of the holes is generally considered: how come the authors did not consider this parameter in the evaluation of drilling performance?

Ad. 10. The Reviewer’s remark is true. The taper of the holes is generally considered. In the current research it was considered too (lines 167-168). The authors do not neglect discussion of this issue. The measurement of the taper was very difficult due to size of the hole (diameter and depth), equipment limitations and blind holes.

It is not very clear what the working current Iw is for the authors? Is it the current discharged in the interelectrode gap? Is it the maximum expected current related to the generated pulse? Please clarify.

Ad. 11. The working current meaning was introduced to the text original text. Just to remind: the threshold current (Ic) is the minimum current value acceptable for feed regulator that should be maintained to support the discharges, while the working current (Iw) is the current value that is reference for feed regulator to be default current during the drilling.

What kind of pulse generator the authors used in the experiments? Transistor or relaxation type? It is important to understand the micro-EDM drilling pulse shape when estimating the performance. Unfortunately, by saying “rectangular voltage pulses” the authors did not help in clarifying this important aspect of the machining.

Ad. 12. Authors included the information in the abstract of the original text that the isoenergetic transistor voltage pulse generator enabling rectangular pulses was applied believing that it would be clear for the Readers reading the full text. Nevertheless this information was added in the full text too according to Reviewer’s 3 remark. The RLC generators generally introduce sinusoidal pulse shape. Maybe the symmetrical pulse on and pulse off times were misleading. Missing word “transistor” was added.

Line 101-102: the authors stated “The chosen polarity of the electrodes ensure maximal material removal rate 101 and minimal electrode tool wear.” What is the polarity of the tool? It is supposed to be negative, but the authors must indicate it clearly.

Ad. 13. Preliminary machining tests revealed excessive tungsten electrode tool wear for negative electrical potential of the machined part. In the current research the positive (higher) electrical potential was applied to the machined part. Authors agree with the Reviewer’s remark that this information should be indicated clearly. Additional information was introduced.

Line 105: Why was the tool not in rotation during the machining of the first Two holes

Ad. 14. The tool was not in rotation during the machining of the first two holes due to sinking approach. Rotation was introduced as a correction of the process in order to improve process’ conditions in the inter-electrode gap.

The quality of Figure 8 is very low: it is very difficult to read the legends and as such to estimate the plots. It is not clear what the x-axis refers to.

Ad. 15. The clarity of the Figure 8 was improved. The x-axis refers to Iw (working current changes).

Line 147 states “in figure 8 the pulse time on and off has…”. please rephrase the sentence “in figure 8, pulse on time and pulse off time have…”.

Ad. 16. Changes were introduced.

Concerning figure 10: A mistake is present in the title of figure related to MRR (pulse on/off vs MMR). Same mistake is present in Table 5.

Ad. 17. Changes were introduced.

Figure 9 and 10 are not commented in the manuscript.

Ad. 18. Figures 9 and 10 are commented in the manuscript now. Changes are yellow highlighted.

Moreover, the authors should explain from a physical viewpoint the following statement: a) “toff and ton have significant impact on hole depth for higher working current values…”.

Ad. 19a. An electrical current intensity is proportional to the number of charges that are transferred in a given time. As electrical potential in this case is constant and the power P=U*I, the bigger I value, the bigger P value. P stands for the energy (work) necessary for material melting and evaporation over time of machining. As the total machining time in this case is constant, the higher P value, the deeper hole depth. Energy in this case is consumed in drilling direction due to regulator compensating interelectrode gap thickness that enables discharges, while side gap thickness is constant due to continuous working electrode feed and circumferential wear. For higher working current values also higher linear electrode wears and hole diameters are observed it makes higher side gap size that can help evacuate debris form interelectrode volume during the pulse off time, that can improve electrical erosion conditions.

b) “…while the increase in hole diameter is negligible”.

Ad. 19b. Yes, the hole diameter is ca. 420µm, while SD=10µm. The smallest diameter was 402 the bigger 450µm. Considering roughness Ra 5-10µm, we believe that this increase could be neglected depending on required tolerance.

From line 149, the authors came back discussing SEM images. This organization is quite confusing. Please, take care of the manuscript structure!

Ad. 20. The reference to Figure 7 was moved to the SEM images section.

In line 168, the authors finally stated that the micro-holes are blind. I warmly suggest to provide this information in the very beginning of the manuscript (title, abstract and so on).

Ad. 21. The information about the holes was moved as suggested.

Line 169: the authors mentioned tool wear, but no measurements or calculations have been done throughout the entire manuscript. This is quite confusing. In line 170, it is stated “the linear wear was measured and presented in the table 5”. It is by no means clear what the authors referred to. Tool wear and linear wear are not the same thing. I strongly suggest to include a definition of the quantity the authors dealt with. In literature, tool wear is often indicated as "volume" tool wear, whereas the linear tool wear refers to the shortening of the tool, assuming that the tool section is kept constant during a micro-EDM process. Nonetheless, micro-EDM drilling of micro-holes is far acknowledged as a process whose tool wear is dramatically outstanding, especially concerning the tool section. It is not just the tool shortening (linear tool wear) but the section variation that matters most, since it introduced the hole tapering. The hole tapering is particularly important in high aspect ratio holes and, although it cannot be easily measured in case of blind holes, it should be extrapolated by the observation of the volume tool wear.

Ad. 22. Authors agree with the Reviewer 3. The words “linear electrode tool” were added and are meant as a electrode tool shortening measured in µm, in order to distinguish linear electrode tool wear and circumferential electrode tool wear. Both electrode tool wears have impact to volumetric electrode tool wear.

Round 2

Reviewer 1 Report

The authors addressed most of the reviewers comments and the manuscript improved, however, the literature still needs update, and the discussion of the results could also be strengthen. Minor English mistakes should also be addressed.   

Author Response

Dear Reviewers,

Once again thank you for your valuable comments and thoughtful immersion in our manuscript. We do hope that our corrections, clarifications and improved discussion based on your remarks would make our manuscript easy and clear to understand by readers. Kindly please find attached our answers to your revisions.

Yours faithfully,

Dominik Wyszynski

Reviewer 3 Report

Although the authors have shown efforts in improving their manuscript and clarify and address the issues I raised previously, some changes are still required to the entire work. I suggest to be more thorough and detailed about definition, acronym and performance indexes, so that the reader can follow and evaluate the results clearly and immediately. I found that the paper is worsen about English clarity in this last version. Finally, it is still ambiguous the statement about the “cost effective and versatile approach to forming c-BN parts using micro EDM”. Please, provide effectiveness to this goal. In the present form is still debatable. 

Other comments the authors are required to address:

First of all, please check typos, grammar and sentence construction.

In particular, please check:

from line 172-177. The sentences are not clear at all.

Furthermore:

Line 27: please replace difficult-to-cut materials (recalled 3 times consecutively in 4 lines) with “category of materials”

Line 79: “some range of the most important parameters set” should be rephrased as follows “ the range of the most important parameters”

Line 79: “was choses”, I guess is “chosen”

Factorial design: what do the authors mean with factorial design here? A factorial design foresees a number of trials and replica according to a design of experiment. The mere variation of some parameters without expressing any combination used for the specific trial does not provide a factorial design. Please explain and address.

Table 1: in relation to pulse frequency. The authors includes two values, but they did not specify whether they use 500 kHz and 50 kHz. When using 500 kHz, the micro EDM is performed in very short pulse regime; When using 50 kHz, the regime is namely "long pulses". As T = ton +toff and F are strictly related to each other, the authors should explicit the settings used for the 6 trials made for the machining of all blind holes. Additionally, such different machining conditions lead to very different results. If the frequency of the pulse is short, more than one pulse can be observed in a time period  T. In this way, the energetic contribution of micro-EDM pulses changes dramatically if you use 50 kHz or 500kHz. Please justify your choices and machining results more clearly in relation to the different machining conditions.

Table 3: I suppose 0.3 is Ic. Unfortunately,  It is reported in table 3, not Ic. This is quite misleading. Please correct.

Acronym: I stated in the previous review that MMR is incorrect. MMR does not stand for Material Removal Rate, MRR is correct. I found the same mistake kept unvaried throughout the manuscript for the second time.

Line 118: about the two trials using sinking approach. It is not clear yet why the authors considered sinking approach if the paper should have "drilling" as main objective. And this fact raises a further doubt about the six trials: are all of them performed using drilling or just 4 of them are drilled? Please clarify.

The introduced paragraph from line 165 to 180 is quite confusing.  Moreover, the sentence part from line 174-175 is by no means clear. 

Line 177: "figure 9 shows" (s is missing).

Line 178: “Iw changes”, I think should be better “Iw variation”. About the “mean drilling speed”: it is not "mean", it is "average drilling speed". The authors should also specify how they consider and calculate the average speed. Please also correct caption of figure 9.

Line 182-194: check format and font size, please.

Table 5: correct MMR with MRR. Moreover, what is the "relative electrode wear"? How do the authors calculate this relative quantity? Details about all indexes cited in the table must be provided in the text, as well. Same for “relative roundness”: it is not clear how the authors calculate this percentage.

Author Response

(The authors gave the same response as above.)
